# Synthesis and Application of ZSM-5 Catalyst Supported with Zinc and/or Nickel in the Conversion of Pyrolytic Gases from Recycled Polypropylene and Polystyrene Mixtures under Hydrogen Atmosphere

**DOI:** 10.3390/polym15163329

**Published:** 2023-08-08

**Authors:** Diego Barzallo, Rafael Lazo, Carlos Medina, Carlos Guashpa, Carla Tacuri, Paúl Palmay

**Affiliations:** 1Facultad Ciencias e Ingeniería, Universidad Estatal de Milagro, Milagro 091050, Ecuador; rlazos@unemi.edu.ec; 2Department of Chemistry, University of Balearic Islands, Cra. Valldemossa 7.5 km, 07122 Palma de Mallorca, Spain; 3Facultad de Ciencias, Escuela Superior Politécnica de Chimborazo ESPOCH, Panamericana Sur Km 1 1/2, Riobamba 060155, Ecuador; carlos.medinas@espoch.edu.ec (C.M.); carlos.guashpa@espoch.edu.ec (C.G.); carla.tacuri@espoch.edu.ec (C.T.)

**Keywords:** catalyst, ZSM-5, monometallic, bimetallic, catalyst acidity, catalyst properties

## Abstract

Currently, catalytic pyrolysis has become a versatile and highly useful technology in the treatment of different plastic wastes. Thus, the development of selective catalysts to carry out cracking reactions and obtain a greater fraction of the desired products is essential. This study focuses on the synthesis of monometallic (Ni) and bimetallic (Ni-Zn) catalysts supported on ZSM-5 zeolite using an impregnation and co-impregnation method, respectively. The obtained catalysts were characterized by FTIR spectroscopy, N_2_ adsorption/desorption measurements, scanning electron microscopy (SEM) and energy dispersive X-Ray spectroscopy (EDS), temperature programmed NH_3_ desorption (TPD-NH_3_) and thermogravimetric analysis (TGA). In this way, a mixture of polystyrene and polypropylene recycled with a catalyst/plastic waste ratio of 1:500 was used for pyrolysis tests. The best results were obtained using the Ni-Zn/ZSM-5 catalyst, which included better impregnation, increased surface acidity, decreased dispersion and a shorter reaction time in the catalytic pyrolysis process. Under the optimized conditions, catalytic pyrolysis showed an excellent performance to generate hydrocarbons of greater industrial interest.

## 1. Introduction

According to review articles reported in the literature, plastic waste treatment [1] promotes the development and implementation of new sustainable environmental technologies for the management of waste, which, due to its high human production worldwide, is usually deposited in dumps, sanitary landfills, and emerging cells as final disposal, generating a great negative impact on the environment. The use of other alternatives including recycling, mechanical and chemical reuse [2], and more efficient technologies, i.e., gasification [3] and pyrolysis [4], provide better solutions to this problem. Pyrolysis involves the high-temperature anaerobic decomposition of plastic waste with or without the presence of a catalyst to obtain biofuels with high calorific values comparable to commercial fuels, generating the development of a circular economy [5]. Moreover, this process has the potential to reduce the carbon footprint of plastic products by reducing carbon dioxide and carbon monoxide emissions [6]. Previous works investigating thermal [7,8] and catalytic [9] pyrolysis have shown its potential, as well as the optimal operating conditions for the recovery of cleaner energy from plastic waste [6,10] or specific mixtures [7]. However, there is currently great interest, especially in catalytic pyrolysis, due to its great advantages, which include the following: (i) it accelerates the reaction rate of the process, allowing a greater depolymerization of the raw material; (ii) it reduces the energy necessary for the degradation of plastic waste; and (iii) it provides greater selectivity to obtain higher-quality pyrolytic products [11]. Nowadays, the catalytic pyrolysis of mixtures of plastic waste is of great interest, because it minimizes the formation of coke and increases the yield and selectivity in obtaining higher-value products, i.e., gasoline and aromatic fractions. However, its selectivity also depends on the type of pyrolysis that is performed [12].

Among the catalysts used in this process are recycled catalysts from fluid catalytic cracking (FCC) units [9], red mud [13], ruthenium homogeneous catalysts [14], nanocatalysts [15] and natural or synthetic zeolites [16,17]. Many researchers have shown that zeolites present a greater potential due to their acidic properties, unique pore size, and large surface area, and they are much more active in the catalytic pyrolysis of different thermoplastics such as polypropylene (PP), polystyrene (PS), high-density polyethylene (HDPE), low-density polyethylene (LDPE) and polyvinyl chloride (PVC) to obtain aromatic hydrocarbons [18,19]. ZSM-5, compared to natural zeolite, has been shown to be more effective as a catalyst in catalytic pyrolysis from plastic waste due to its more ordered structure and smaller pore size, which provides greater selectivity and catalytic activity, increasing the efficiency of the conversion of thermoplastics into valuable products. These products can be used as fuels or raw materials for the production of chemical products, i.e., solvents, resins, and other polymeric materials [20]. Moreover, ZSM-5 presents high cracking activity and low coke production and contains impurities such as solid residues, i.e., nitrogen and phosphorus [13,19].

Generally, the introduction of metal species in ZSM-5, such as Mo, Zn, Ag, Ni, Cu, Sn, either on the external surface or in the framework, is a frequently used method to improve catalytic activity. Transition metal-supported catalysts can be synthesized through various methods, including ion exchange, incipient wetness impregnation, or the deposition of metal complexes, followed by calcination or reduction to achieve oxidized states or metal nanoparticles. However, the impregnation method is the most used to immobilize acids and bases, salts, oxides, or complexes on oxide supports. In the case of zeolite support, the active transition metal can be incorporated using ion exchange and subsequent processing to produce materials with appropriate porous and chemical properties [21]. In this way, the incorporation of transition metals onto natural zeolite (Ni/Z) [22] or Ni/ZSM-5 [23] improves its catalytic properties, increasing its selectivity for the production of short-chain biofuels, for its industrial application [24]. Iliopoulou et al. [25] found that the presence of NiO and Co3O4 nanoparticles supported on ZSM-5 positively impacts the deoxygenation process, and improves the pyrolysis oil yield with minimal coke formation. In addition, bimetallic impregnation such as Co-Mo/Z [22] and Zn-Ni/ZSM-5 [26], have been shown to provide greater stability and catalytic performance, which is very useful to reduce energy input to carry out pyrolysis. Miskolczi et al. [27] studied the impregnation of metals, including Ni and Zn, on the surface of ZSM-5 and y-zeolite catalysts for the thermocatalytic pyrolysis of plastic waste, and showed a greater decrease in the activation energy with Ni compared to Zn. It also increased the hydrothermal stability of the catalyst due to its dehydrogenating activity and its moderate acid strength. In any case, both catalysts maximize the yield of aromatic compounds. In addition, Lu Wang et al. [28] presented evidence of the potential of the Zn/ZSM-5 catalyst by reducing the amounts of oxygenated compounds through deoxygenation reactions, which allows one to obtain a higher concentration of hydrogen and aromatic compounds compared to individual ZSM-5. Moreover, Stanton et al. [29] indicated that the cofeeding of H_2_ together with biomass in the pyrolysis process allows one to obtain products with a lower oxygen content and with reduced coke formation, which blocks the pores and acid sites of the catalyst, causing its rapid deactivation. In this sense, all metal-modified ZSM-5 catalysts (Cu, Ga, Ni, Co, Pt) produced fewer oxygenate compounds during pyrolysis with hydrogen feed compared to without hydrogen addition; however, Ni-ZSM-5 exhibited a higher hydrogenation activity, resulting in fewer polyaromatics progressing to coke [30]. In addition, Ni/ZSM-5 showed an increase in the yield of hydrocarbons obtained with hydrogen addition comparable to unmodified ZSM-5.

Thus, the main objective of this study is to evaluate the pyrolytic products obtained through GC-MS from mixtures of polypropylene and polystyrene plastic waste using two types of synthesized catalysts, either a single-metal (Ni/ZSM-5) or bimetallic (Zn-Ni/ZSM-5) compound, deposited on zeolite by means of the impregnation/co-impregnation method, respectively, with the external addition of hydrogen. In addition, different characterization tests were carried out with both catalysts to demonstrate their physicochemical properties, which included their morphology, surface area, acid properties, and composition. Moreover, the results obtained were compared without catalyst under a hydrogen atmosphere.

## 2. Materials and Methods

### 2.1. Sampling

The sampling of individual polypropylene and polystyrene plastic waste was carried out completely randomly for ten days over three months, collecting 5 kg/day. The plastic waste was obtained from a dump located in the city of Riobamba, Ecuador, a city with a population of approximately 250,000 habitants. In the first month they were collected during the first ten days, in the second month during the intermediate days and in the third month during the last ten days. Subsequently, these residues were ground to a size of 1 cm and washed with 1% sodium hydroxide to remove interferents and gums. Subsequently, these residues were characterized through FTIR using a JASCO FT/IR-4100 spectrometer (PerkinElmer, Waltham, MA, USA).

### 2.2. Preparation of Zeolite ZSM-5

Zeolite ZSM-5 was purchased from Sigma Aldrich (St. Louis, MO, USA). A weight of 100 g of ZSM-5 was mixed with 400 mL of ammonium nitrate (1 mol L^−1^) (Fisher, Polk County, MN, USA) and the mixture was heated at 80 °C for 1 h using a rotary evaporator (Biobase model RE100-Pro) with continuous stirring at 60 rpm, which allowed us to increase the protonation of the zeolite before its impregnation. After that, several washes with distilled water were performed and excess solvent was removed by vacuum filtration. The obtained solid was dried at 120 °C for 2 h and then consecutively calcined at 250 °C for one hour, at 400 °C for an additional hour and finally for 12 h at 500 °C.

### 2.3. Synthesis of Ni/ZSM-5 and Zn-Ni/ZSM-5 Catalysts

For the synthesis of the monometallic (Ni/ZSM-5) and bimetallic (Zn-Ni/ZSM-5) catalyst, the impregnation and co-impregnation method were used, respectively. For both syntheses, 50 g of protonated ZSM-5 was mixed with 100 mL of ammonium nitrate solution (2 mol L^−1^) and heated at 80 °C for 1 h using a rotary evaporator with continuous stirring at 60 rpm. Then, 100 mL of nickel nitrate solution (1 mol L^−1^) was added and stirred for an additional hour. The bimetallic synthesis was carried out in a Ni-Zn ratio (1:1, *v*/*v*); thus, 100 mL of zinc nitrate solution (1 mol L^−1^) (Alfa Aesar, Haverhill, MA, USA) was added to the above mixture and stirred for an additional hour. For both syntheses, the solid formed was washed three times with methanol and separated using vacuum filtration. The solid obtained was dried at 120 °C for 2 h for subsequent calcination at 500 °C for 7 h.

### 2.4. Catalyst Characterization

The morphology and particle size of the samples were characterized using a scanning electron microscope (JEOL model JSM-IT100) coupled to an energy-dispersive spectrometer (SEM-EDS). The identification of the functional groups present in the plastic waste as well as the chemical modification of the surface of the synthesized catalysts were performed using Fourier transform infrared (FTIR) spectroscopy using a JASCO FT/IR-4100 spectrometer. Nitrogen adsorption–desorption isotherms were acquired at 63 K with a flow of 50.3 cm^3^ using an AutoChem II 2920 gas-adsorption instrument. The data obtained were analyzed using the Brunauer–Emmett–Teller (BET) model to obtain the superficial area. Thermogravimetric analysis (TGA) was performed using the Perkin-Elmer TGA7 equipment. Finally, the surface acidity of the samples was determined using the ammonia desorption technique, with a flow of 20.21 cm^3^ min^−1^ at a programmed temperature (TPD-NH_3_) up to a temperature of 400 °C, and using the adsorption–desorption technique of hydrogen at a flow of 20.09 cm^3^ min^−1^ (TPD-H_2_ and TPR-H_2_), which was carried out at a maximum programmed temperature of 500 °C.

### 2.5. Pyrolysis Conditions

The pyrolysis experiments were carried out in the GSH-5.0L reactor (Weihai Global Chemical Machinery, Weihai, China), as shown in Figure 1. This proposed system consisted of a stirred reactor, which has heated by electrical resistance and was monitored by means of an internal PID temperature control. The reactor outlet was connected to a rectification column, where pyrolytic gases circulated, then passed to a condensation system that had a cold-water inlet at 10 °C from a chiller. For catalytic pyrolysis, the rectification column previously used in thermal pyrolysis was modified to convert it into a fluidized bed tubular reactor, with a catalyst/plastic waste ratio of 1:500, which was used for pyrolysis tests and through which the pyrolytic gases circulated, maintaining their temperature at 250 °C. In addition, to linearize the polymeric structure of the plastic waste, hydrogen gas was injected with a flow of 1 L min^−1^.

For each test, 1000 g of the PS (75%) + PP (25%) plastic mixture was fed to the reactor and heated up to a temperature of 450 °C with a heating rate of 15 °C min^−1^, at a pressure of 0.1 MPa, with stirring at 20 rpm and the addition of external hydrogen at 1 L min^−1^. The liquid fraction obtained from the condensation of hydrocarbons was collected in amber bottles and stored at a low temperature (<4 °C) to avoid their volatilization. Subsequently, these samples were analyzed by gas chromatography with mass spectrometry (GC-MS) with a detector temperature of 250 °C, using helium as carrier gas at 1 mL min^−1^.

## 3. Results

### 3.1. Characterization of Polypropylene and Polystyrene Waste

FTIR has been used for the chemical characterization of both plastic wastes, i.e., polypropylene (PP) and polystyrene (PS). The PP spectrum (Figure 2a) shows three groups of bands, which correspond to the tension movements of the CH bonds at 2900 cm^−1^, C-C tension movements at 1350–1470 cm^−1^, and a bending movement of -CH_3_ between 1200 and 1000 cm^−1^. The PS spectrum (Figure 2b) presents three groups of absorption bands, which shows the multiple tension movements of the C-H bonds at 2700–3000 cm^−1^, C-C at 1400–1600 cm^−1^, of the aromatic ring and a bending movement of -CH_2_ and tensions of aromatic rings between 700–800 cm^−1^. In addition, both spectra obtained coincide with the results obtained in other works reported in the literature [6].

### 3.2. Physicochemical Characterization of Ni/ZSM-5 and Ni-Zn/ZSM-5

The catalysts obtained were characterized using FTIR spectroscopy (Figure 3). The IR spectrum of the Ni/ZSM-5 catalyst shows four absorption bands at 998, 813, 674, 570 cm^−1^, which match well with those of Ni/ZMS-5 previously reported by other authors [24]. The band at 998 cm^−1^ corresponds to the asymmetric stress vibration of Si-O-Si, while the band at 813 cm^−1^ is attributed to the symmetric stress vibration of the tetrahedrons silicon and aluminum. The band at 674 cm^−1^ is assigned to the internal bending vibrations of the five-membered rings of T-O-T (T = Si or Al) on the ZSM-5 structure. The last peak of 570 cm^−1^ corresponds to the vibration caused by the double rings of four members of the outer bonds [26]. The FTIR spectrum of the Ni-Zn/ZSM-5 shows similar absorption bands to those of the Ni/ZSM-5, ranging from 1002 to 563 cm^−1^. In both cases, the band at 563 cm^−1^ is assigned to bond torsion vibrations between Al-O-Si and the presence of vibrations of Ni or Zn secondary units on the ZSM-5 zeolite crystalline structure [23], confirming their incorporating on the zeolite. The IR spectrum of the bimetallic catalyst exhibits absorption bands with less intensity, indicating the scarce co-impregnation of Ni and Zn simultaneously, since both transition metals compete for their incorporation on the zeolite structure, limiting sintering in equal proportions [27].

The morphology and crystal size of the prepared materials (Ni/ZSM-5 and Ni-Zn/ZSM-5) were investigated by SEM analysis. Figure 4a,b shows the SEM micrograph of the Ni/ZSM-5 (monometallic catalyst) and Ni-Zn/ZSM-5 (bimetallic catalyst), respectively. In both images, a tetragonal shape of ZSM-5 zeolites after Zn and/or Ni modification was evidenced, which was in agreement with the typical form of parent ZSM-5, indicating that both remain unchanged after the functionalization process [31]. As can be seen in the micrographs, both materials are formed from agglomerates of particles on ZSM-5, with a large crystal size of about 800 nm. In addition, with Ni/ZSM-5, the particle size of the zeolite aggregates increased from 5 µm to 10 µm; thus, a greater spatial distribution of particles and a small fragment of agglomerates were obtained with Ni-Zn/ZSM-5 [24].

Additionally, energy-dispersive X-ray spectroscopy (EDX) of the synthesized catalysts was performed. The EDS spectra show the elemental composition of the monometallic and bimetallic catalyst, confirming the presence of zinc and/or nickel (Figure 4c). It shows five characteristic peaks of O, Ni, Na, Al and Si located at 0.52, 0.90, 1.03, 1.47 and 1.72 keV, respectively. Figure 4d reveals the presence of zinc on ZSM-5, due to the partial cation exchange that occurred instead of Na, which is present in its structure as a charge compensator in aluminum tetrahedrons (M + AlO_3_) [27]. The weight percentage of major components were found to be C (33.0%), O (47.8), Al (7.4%) y Si (6.3%). The minor components included Ni (2.5%) and Zn (3.0%), which, in agreement with the literature, have a higher affinity for Zn on the ZSM-5 zeolite and Si/Al ratio, indicating a very similar distribution on all ZSM-5 zeolites [28]. However, it is important to note that the results obtained can vary from sample to sample. This variability arises because the spectrum is acquired from a single point, which may not always be fully representative of the entire sample. Thus, for a more accurate quantitative elemental analysis, utilizing techniques such as inductively coupled plasma (ICP) and X-ray fluorescence (XRF) is recommended [32]. These methods provide precise measurements of elemental concentrations and are well-suited for determining the atomic ratios of Ni/Zn.

The textual properties of the materials were analyzed by N_2_ adsorption/desorption measurements, as shown in Table 1. The BET-specific surface and total pore volume of the Ni/ZSM-5 were 1.80 m^2^ g^−1^ and 0.41 cm^3^ g^−1^, respectively, which are comparable to those reported in the literature [32]. However, the values of the surface area and the total volume pore for Ni-Zn/ZSM-5 decreased to 1.38 m^2^ g^−1^ and 0.32 cm^3^ g^−1^, respectively, which is probably due to the impregnation of Zn on the pores of the ZSM-5 zeolite. After the incorporation of Ni and Zn onto the ZSM-5 structure, a mesoporic structure was obtained, which was corroborated by the findings of Wei et al. [31], indicating that unmodified ZSM-5 presents microporous structures, and after the incorporation of the metals mesopore structures were created. In any case, the pore volume of ZSM-5 was smaller than modified zeolites (monometallic and bimetallic catalyst), which can be attributed to an impregnated metal support layer, causing structural modifications on the surface of the zeolite [33,34] and a change in charge of the metal [25]. Therefore, this increase facilitates the separation of oxygen compounds and inhibits the superficial polymerization of ZSM-5, improving the selectivity of aromatic hydrocarbons in catalytic reactions and reducing coke formation. In addition, both catalysts have a pore size distribution in the range 3–10 nm.

The acidity of the catalysts by TPD-NH_3_ allowed to determine the total density of acid sites. Thus, as the number of acid centers increases, the quantity of adsorbed ammonium also increases. According to the reported acidity values, it was shown that Ni-Zn/ZSM-5 has a higher acidity (0.626 mmol·g^−1^) than Ni/ZSM-5 (0.187 mmol·g^−1^) and ZSM-5 (0.103 mmol·g^−1^). Thus, both catalysts show higher acidity compared to ZSM-5 (Table 1). This behavior can be explained by the cation prevalence, especially in the bimetallic catalyst [35].

The results obtained for hydrogen desorption show that the TPD-H_2_ for Ni/ZSM-5 was 0.0252 mmol·g^−1^, while for Ni-Zn/ZSM-5 it was 0.0011 mmol·g^−1^. The monometallic catalyst exhibits a higher value compared to the bimetallic catalyst, despite both having relatively low values, due to the low presence of nickel and zinc crystals in the catalysts. The hydrogen desorption occurs at a temperature of 500 °C. On the other hand, the results obtained for metallic dispersion are low, which can be attributed to the presence of Ni-Zn metallic loads in the catalysts, affecting the dispersion percentage. In the bimetallic catalyst, it is evident that the co-impregnation of zinc leads to a decrease in the hydrogen chemisorption capacity compared to the monometallic one, resulting in a decrease in metallic dispersion from 0.0657% to 0.0298%. Moreover, the addition of zinc increases the size of the nickel particles, consequently decreasing the metallic dispersion [36]. The chemisorption pulses for the Ni/ZSM-5 and Ni-Zn/ZSM-5 catalysts shows that the bimetallic catalyst presents a higher signal compared to the monometallic catalyst, indicating a decrease in hydrogen consumption in the reduced species of the Ni/ZSM-5 catalyst, due to the fact that the nickel may be occluded in the zinc during the calcination process at 500 °C. This is supported by the TPR-H2 results obtained for Ni/ZSM-5 (4.88 cm^3^ g^−1^) and Ni-Zn/ZSM-5 (10.79 cm^3^ g^−1^), where the reduced species of the bimetallic catalyst were significantly higher than those of the monometallic catalyst, indicating a decrease in hydrogen consumption in the reduced species.

Thermogravimetric analysis (TGA) was performed on both catalysts to determine their thermal stability. The Ni-Zn/ZSM-5 catalyst showed a higher mass loss compared to the Ni/ZSM-5 catalyst that included volatile compounds, organic matter and non-volatile material, as shown in Figure 5. During the heating process, zinc oxide species are formed, leading to the observed higher mass loss in the bimetallic catalyst. In this way, both synthesized catalysts exhibit three specific weight-loss regions under thermal treatment, as shown in Figure 6. The first weight loss region below 200 °C is attributed to physiosorbed water molecules, resulting from the hydration effects during the catalyst synthesis stage, the desorption of organic compounds, and the removal of impurities. The second region of weight loss (300–500 °C) corresponded to strongly bound water molecules present in the framework structure. The third region at temperatures above 550 °C can be attributed to the formation of metal oxides and changes in the crystalline structure of the zeolite [36,37].

### 3.3. Studies of Catalytic Activity in Pyrolysis

Figure 6 shows the behavior of the reaction temperature vs. time for the standard test with both the monometallic catalyst and with the bimetallic catalyst. It can be observed that the catalytic action causes a decrease in the time needed to reach the reaction temperature of 450 °C for the reaction without the presence of a catalyst, confirming the decrease in the activation energy caused by the catalytic action. On the other hand, a slight decrease in time can be observed in the curve corresponding to the bimetallic catalyst, at a temperature about 350 °C. This variation can be attributed to the slightly higher acidity of the catalyst, which may have contributed to its enhanced performance.

Figure 6b shows the classification by family of the products identified using gas chromatography–mass spectrometry (GC-MS). It can be observed that when the pyrolysis process of the polystyrene/polypropylene mixture is carried out, the greater presence of PS leads to the formation of an important fraction of aromatics. However, it can be seen that working with Ni-Zn catalysts generates a slight increase in the production of compounds such as aromatics, alkanes and aldehydes, which can be attributed to the attack on the acid sites and the hydrogen exchange between the carbonated chains. Thus, the molecular weight of the chains decreases, with the formation of coke and oligomers reflected in the recovery of the styrene monomer or its isomers. In addition, a decrease in alkenes can be observed with an increase in alkanes, both for the monometallic and bimetallic catalysts, due to the action of the catalyst and the injection of hydrogen. The Ni-Zn catalyst, having a higher number of acid points, facilitates the alkylation between the olefin, giving rise to saturated branched hydrocarbons, with a slight increase in the number of alkanes.

In addition, Table 2 shows the individual components obtained from the pyrolysed oil using the bimetallic catalyst and GC-MS, which were adjusted according to existing references. The constituent components are aromatics, alkanes, alkenes, ketones, cyanates, esters, alcohols, and amines.

## 4. Conclusions

The catalysts synthesized for this research corresponded to heterogeneous catalysts using a ZSM-5 support, with Ni impregnation and Ni-Zn co-impregnation. Studies were been carried out to determine which one has better physical properties. In IR spectroscopy, the bands corresponding to the symmetric and asymmetric stretching vibrations of the Si-O-Al atoms in the two catalysts were detected, as well as bands indicating the presence of secondary units in the zeolite structure (Ni and Zn). It has been possible to confirm the presence of metals in the catalysts through SEM-EDS, where the presence of Ni was found in the monometallic catalyst and the bimetallic Ni-Zn in different proportions, with a quasi-spherical shape and a particle size that did not exceed 5 µm. Moreover, it is known that the two synthesized catalysts have acidity indices within the allowed range for a ZSM-5 zeolite, from which it can be concluded that the bimetallic catalyst has better physicochemical properties. In terms of catalytic activity, both catalysts increase reaction rate by reducing the pyrolysis temperature time, which is due to their acidity, compared to the results obtained without a catalyst under a hydrogen atmosphere. In terms of the products formed, the bimetallic catalyst causes a slight increase in the formation of aromatics, alkanes, and aldehydes, while the monometallic catalyst produces a greater formation of alkenes, ketones, amines, and esters.

However, in future work, other catalyst-characterization techniques could be explored, such as (i) powder X-ray diffraction (PXRD), to determine the crystalline structure and identify the phases present in the sample, and (ii) inductively coupled plasma (ICP), or X-ray fluorescence (XRF), for more accurate quantitative elemental analysis.

## Figures and Tables

**Figure 1 polymers-15-03329-f001:**
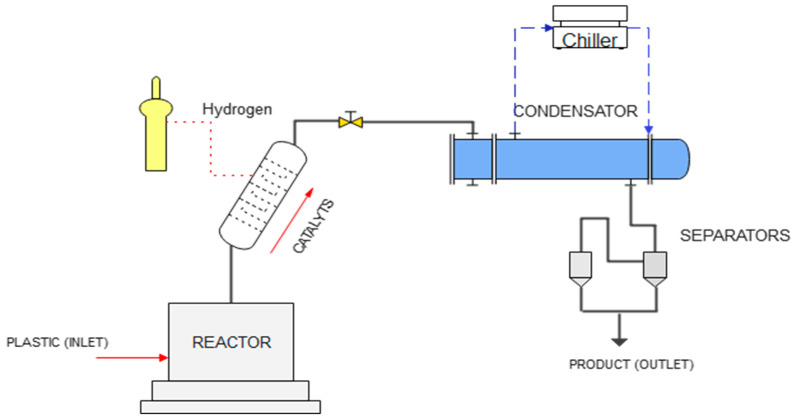
Schematic representation of the pyrolytic reactor.

**Figure 2 polymers-15-03329-f002:**
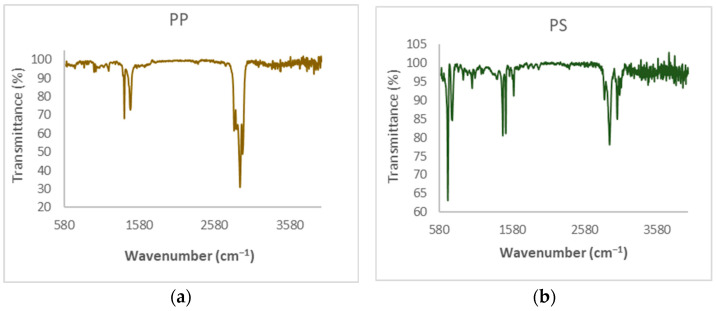
FTIR Spectra of raw: (**a**) polypropylene and (**b**) polystyrene.

**Figure 3 polymers-15-03329-f003:**
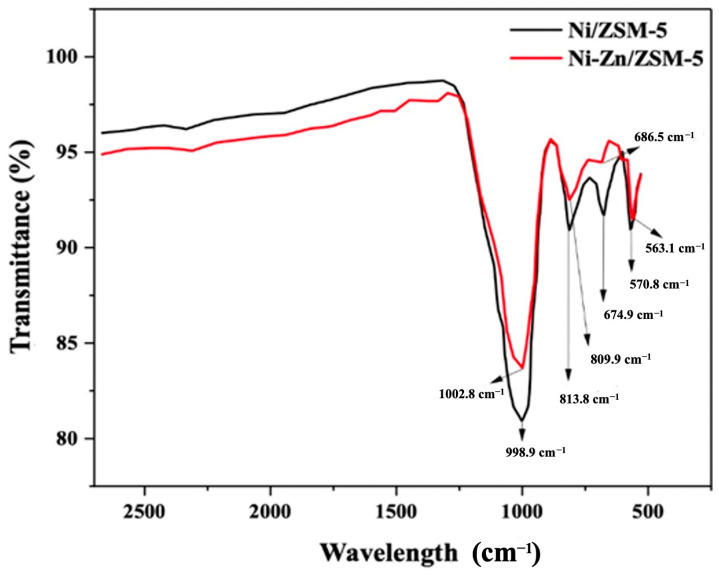
IR spectrum of Ni/ZSM-5 and Ni-Zn/ZSM-5.

**Figure 4 polymers-15-03329-f004:**
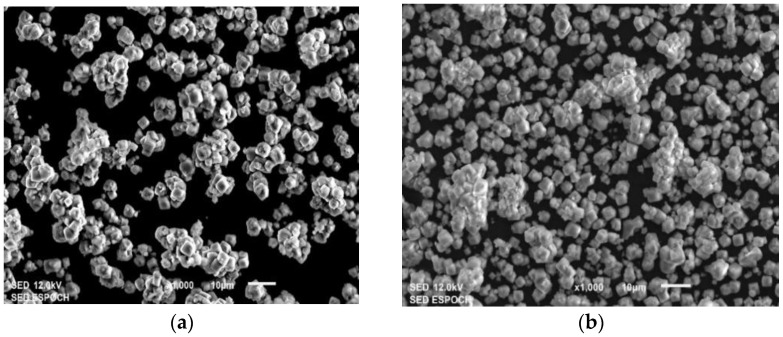
Scanning electron microscopy images of (**a**) Ni/ZSM-5 and (**b**) Ni-Zn/ZSM-5. EDS spectra of (**c**) Ni/ZSM-5 and (**d**) Ni-Zn/ZSM-5.

**Figure 5 polymers-15-03329-f005:**
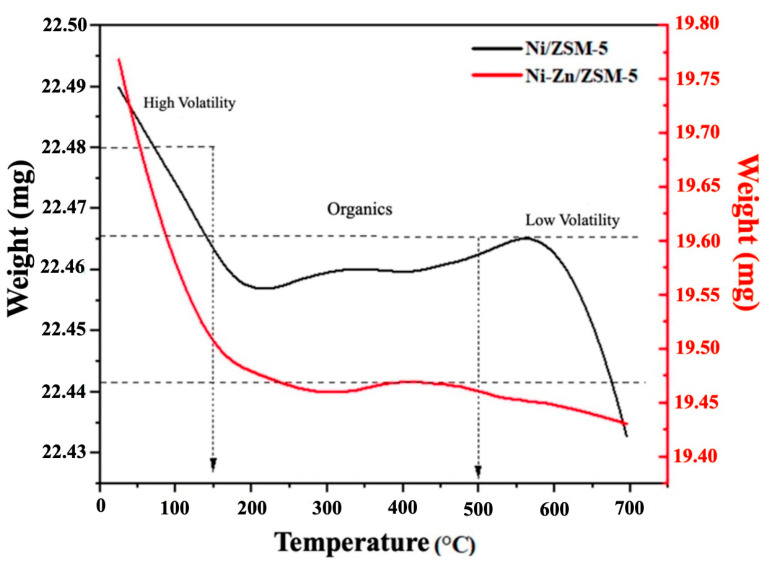
Thermogravimetric curves of Ni/ZSM-5 and Ni-Zn/ZSM-5 catalysts.

**Figure 6 polymers-15-03329-f006:**
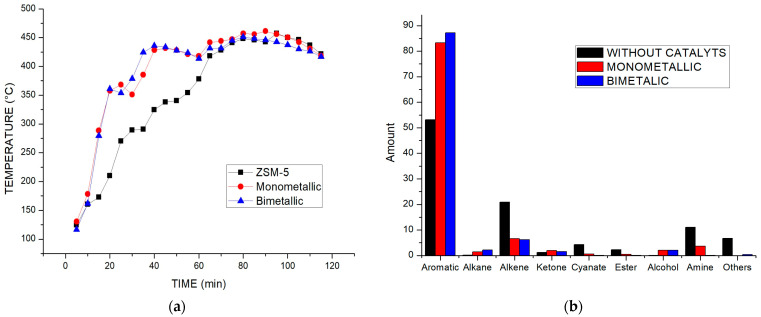
(**a**) Behavior of catalytic pyrolysis reactor temperature with respect to time, using ZSM-5, Ni/ZSM-5 and Ni-Zn/ZSM-5. (**b**) Identified products of catalytic pyrolysis using gas chromatography coupled with mass spectrometry (GC-MS).

**Table 1 polymers-15-03329-t001:** Porosity and acidity characteristics of zeolitic catalysts.

Catalyst	Specific Surface Area m^2^ g^−1^	Pore Volume cm^3^ g^−1^	Number of Acid Sites mmol·g^−1^
ZSM-5	354	0.10	0.103
Ni/ZSM-5	180	0.41	0.187
Ni-Zn/ZSM-5	138	0.32	0.626

**Table 2 polymers-15-03329-t002:** Components of the pyrolysis oil from recycled polypropylene and polystyrene mixtures with Ni-Zn/ZSM-5.

Peak Number in Chromatogram	tr (min)	Compound	% Abundance
1	1.55	Toluene	4.77
2	1.63	Heptane, 2,4-dimethyl-	0.20
3	1.68	2,4-Dimethyl-1-heptene	5.33
4	1.75	Ethylbenzene	7.02
5	1.89	Bicyclo [4.2.0] octa-1,3,5-triene	16.61
6	1.97	Benzene, (1-methylethyl)-	2.56
7	2.09	Benzene, propyl-	0.13
8	2.24	alpha-Methylstyrene	11.11
9	2.36	Tridecane, 4-methyl-	0.13
10	2.38	Heptane, 4-methyl-	0.16
11	2.50	Benzene, 2-propenyl-	0.34
12	2.72	Benzene, (2-methylpropyl)-	0.40
13	2.88	2-Decenal	0.51
14	2.92	Cyclopropane, 1,1-dimethyl-	0.39
15	3.34	4-Pentenal, 2-ethyl-	0.19
16	3.49	Benzene, 3-pentenyl-	0.37
17	5.91	2-Decene, 7-methyl-	0.91
18	6.05	Hexane, 2,3,4-trimethyl-	0.32
19	6.15	Benzene, (3-methylbutyl)-	0.53
20	6.22	Hexane, 2,3,4-trimethyl-	0.65
21	7.08	Benzene, (1-methylenebutyl)-	0.34
22	7.45	Benzene, 3-pentenyl-	0.31
23	10.01	Bibenzyl	0.39
24	10.70	Benzene, 1,1′-(1-methyl-1,2-ethanediyl) bis-	0.43
25	13.01	Benzene, 1,1′-(1,3-propanediyl) bis	15.53
26	13.19	1,2-Diphenylcyclopropane	0.30
27	13.53	p-Xylene	3.02
28	14.37	3-Butynylbenzene	9.09
29	14.52	Ethanone, 2-(2-ethenylphenyl)-1-phenyl-	1.56
30	14.61	1,2-Diphenylcyclopropane	2.64
31	14.81	Cyclohexane, 1,2,4-trimethyl-	0.35
32	14.95	Benzene, 1,1′-(1,4-butanediyl) bis-	0.41
33	15.39	Benzene, 1,1′-(3-methyl-1-propene-1,3-diyl) bis-	0.67
34	15.84	Benzene, 1,1′-(3-methyl-1-propene-1,3-diyl) bis-	0.32
35	16.24	Benzene, 1,1′-(1-butene-1,4-diyl) bis-	0.53
36	16.88	Naphthalene, 1-phenyl-	0.55
37	19.02	Naphthalene, 2-phenyl-	1.02
38	19.34	Spiro (tricyclo [6.2.1.0(2,7)] undeca-2,4,6-triene)-7,1′-cyclopropane	0.44
39	22.25	m-Terphenyl	0.35
40	22.89	p-Terphenyl	0.29
41	27.17	trans-2-Phenyl-1-cyclohexanol	2.1
42	27.31	1-benzyloxy-3-methyl-4H-1,2,4-triazol-3-ol, 5-[(phenylmethyl)	3.28
43	30.89	1,1′:2′,1″-Terphenyl, 4′-phenyl-	1.18
44	34.23	1,1′:3′,1″-Terphenyl, 5′-phenyl-	2.29

## Data Availability

The data presented in this study are available on request from the corresponding author.

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
