# Peer review of "Synthesis and Application of ZSM-5 Catalyst Supported with Zinc and/or Nickel in the Conversion of Pyrolytic Gases from Recycled Polypropylene and Polystyrene Mixtures under Hydrogen Atmosphere"

_polymers, 2023, doi:10.3390/polym15163329_

Round 1

Reviewer 1 Report

The manuscript describes the application of a ZSM-5 catalyst supported with zinc and/or nickel in the conversion of pyrolytic gases. The whole design is not that novel, but all the data and experiments are well displayed. The catalysts were first synthesized by different methods and then characterized. Using Ni-Zn/ZSM-5 catalyst, the pyrolysis shows better impregnation, increased surface acidity, decreased dispersion, and shorter reaction time.

Therefore, I support its publication in this journal.

Author Response

Thanks for your comments

Reviewer 2 Report

In the present manuscript, the authors studied the impact of the modified ZSM-5 catalysts with single metal Zn and bi metallic Ni-Zn. The catalysts synthesized for this research correspond to heterogeneous catalysts using a ZSM-5 support, with Ni impregnation and Ni-Zn co-impregnation. The authors studied determined and characterized using IR spectroscopy. The authors confirmed the presence of metals in the catalysts through SEM-EDS, where the presence of Ni has been found in the monometallic catalyst and the bimetallic Ni-Zn in different proportions with quasi-spherical shape and particle size that does not exceed 5 µm. The acidity indices of the two modified ZSM-5 catalysts were found to be within the range for a ZSM-5 zeolite, from which it is concluded that the bimetallic catalyst has better physicochemical properties. In terms of catalytic activity, both catalysts increase the reaction rate by reducing the pyrolysis temperature time, which is due to their acidity compared to the results obtained without catalyst under hydrogen atmosphere. In terms of the products formed, the bimetallic catalyst produces a slight increase in the formation of aromatics, alkanes, and aldehydes, while the monometallic catalyst produces a greater formation of alkenes, ketones, amines, and esters.

All the above results were impressive and are potential insights to develop efficient catalysts which can employ effective catalytic pyrolysis leading to highly needed resources with high yields. All the characterizations are satisfactory and well presented. Hence it is recommended to publish this manuscript in Polymers as such.

Author Response

Thanks for your appreciation

Reviewer 3 Report

The work of P. Palmay and co-authors are devoted to the current topic of plastics processing. The authors presented the synthesis of bimetallic catalysts for thermal degradation. After reading the manuscript the following comments and questions arose.

In the experimental part "2.3. Synthesis of Ni/ZSM-5 and Zn-Ni/ZSM-5 catalysts" the authors write "50 g of protonated ZSM-5 was dissolved in 100 mL of ammonium nitrate". What does dissolving between two solids mean? The term solubility is not used quite correctly, in my opinion. Further in the same method "Finally, both synthesized catalysts were washed several times, and then vacuum filtration was carried out." The question arises what was washed and in what form was the reaction mass?

Based on the data presented on the catalysts, it is not possible to state that one phase corresponding to one substance and not a mixture was obtained. In that case, PXRD would be appropriate.

On the catalytic part, there is also a question. Figure 7 is poorly readable, especially the caption on it with the classes of organic substances.

The substances analyzed by GC-MS should be presented in more detail in the SI (chromatograms, retention times). The assignment of substances to classes of substances does not give a complete picture and does not allow to compare catalysts on efficiency.

Author Response

We would like to thank Reviewer 3 for its kind questions and comments to improve the paper.

Please find next the reply to its requirements.

  • In the experimental part "2.3. Synthesis of Ni/ZSM-5 and Zn-Ni/ZSM-5 catalysts" the authors write "50 g of protonated ZSM-5 was dissolved in 100 mL of ammonium nitrate". What does dissolving between two solids mean? The term solubility is not used quite correctly, in my opinion.

It has been modified by: “50 g of protonated ZSM-5 was mixed with 100 mL of ammonium nitrate solution (2 mol L-1)”.

  • Further in the same method "Finally, both synthesized catalysts were washed several times, and then vacuum filtration was carried out." The question arises what was washed and in what form was the reaction mass?

It has been modified by: “For both syntheses, the solid formed was washed three times with methanol and separated using vacuum filtration”

  • Based on the data presented on the catalysts, it is not possible to state that one phase corresponding to one substance and not a mixture was obtained. In that case, PXRD would be appropriate.

Thanks for your appreciation. It has been included: “However, in future work, other catalyst characterization techniques could be explored, such as: i) Powder X-ray diffraction (PXRD) to determine the crystalline structure and identify the phases present in the sample and ii) Inductively Coupled Plasma (ICP) or X-ray Fluorescence (XRF) for more accurate quantitative elemental analysis”.

  • On the catalytic part, there is also a question. Figure 7 is poorly readable, especially the caption on it with the classes of organic substances.

This has been modified

  • The substances analyzed by GC-MS should be presented in more detail in the SI (chromatograms, retention times). The assignment of substances to classes of substances does not give a complete picture and does not allow to compare catalysts on efficiency.

Reply: Table 2 has been included with individual components of the pyrolisis oil from recycled polypropylene and polystyrene mixtures with Ni-Zn/ZSM-5

Reviewer 4 Report

1) Authors claimed that the binding energies for both Ni and Zn were at 0.90keV. How can they be identified? Please comment and get compositional analysis on the atomic ratio of Ni/Zn as compared to the starting ratio.

2) Table 1: The pore volume of ZSM-5 is 0..10 cm4 g-1, which is lower than Ni/ZSM-5 and Ni-Zn/ZSM-5. Please explain

3) Both Ni/ZSM-5 and Zn-Ni/ZSM-5 were calcined at 500C for 7hr as described in experimental part. Please explain why there was weight loss before 500C  in Figure 6.

4) Figure 7a: The data for ZSM-5 is missing. The authors mentioned the reactor was heated up to a temperature of 450C with a heating rate of 15C min-1. However, the results show the reactor reaches 450C in 70min. Please explain the inconsistency.

5) The authors claimed that the  catalytic action causes a decrease in the time needed to reach the reaction temperature of 450° C for the reaction without the presence of a catalyst. On the other hand, a slight decrease in time can be observed in the curve corresponding to the bimetallic catalyst. This can be attributed to the slightly higher acidity of the bimetallic catalyst. This is not true in Figure 7a. Also the temperatures  at 0 min was different for two cases (100C and 125C for monometalic and bimetalic one, respectively).

6) Please improve the quality of the figures (3, 6 and 7) which are blurred 

Moderate editing of English language is needed.

Author Response

We would like to thank Reviewer 4 for its kind questions and comments to improve the paper.

Please find next the reply to its requirements.

1) Authors claimed that the binding energies for both Ni and Zn were at 0.90keV. How can they be identified? Please comment and get compositional analysis on the atomic ratio of Ni/Zn as compared to the starting ratio.

Reply: It has been modificated.

It has been modified by: “The EDS spectra show the elemental composition of the monometallic and bimetallic catalyst, confirming the presence of zinc and/or nickel (Figure 4c). It shows five characteristic peaks of O, Ni, Na, Al and Si located at 0.52, 0.90, 1.03, 1.47 and 1.72 keV, respectively. Figure 4d reveals the presence of zinc on ZSM-5, due to the partial cation exchange that occurred instead of Na, which is present in its structure as a charge compensator in aluminum tetrahedrons (M+AlO3) [34]. The weight percentage of major components were found to be C (33.0%), O (47.8), Al (7.4%) y Si (6.3%). The minor components included Ni (2.5%) and Zn (3.0%), which, in agreement with the literature, there is a higher affinity for Zn on the ZSM-5 zeolite and Si/Al ratio, indicating a very similar distribution on all ZSM-5 zeolites [26]. However, it is important to note that the results obtained can vary from sample to sample. This variability arises because the spectrum is acquired from a single point, which may not always be fully representative of the entire sample. Thus, for a more accurate quantitative elemental analysis, it is recommended to utilize techniques such as inductively coupled plasma (ICP) and X-ray fluorescence (XRF) [34]. These methods provide precise measurements of elemental concentrations and are well-suited for determining the atomic ratios of Ni/Zn”.

2) Table 1: The pore volume of ZSM-5 is 0.10 cm4 g-1, which is lower than Ni/ZSM-5 and Ni-Zn/ZSM-5. Please explain

Reply: It has been modificated.

This has been included: “The pore volume of ZSM-5 is smaller than modified zeolites (monometallic and bimetallic catalyst), which can be attributed to impregnated metal support layer, causing structural modifications on the surface of the zeolite [36,37] and charge of the metal [38]. Therefore, this increase facilitates the separation of oxygen compounds and inhibits the superficial polymerization of ZSM-5, improving the selectivity of aromatic hydrocarbons in catalytic reactions and reducing coke formation”.

3) Both Ni/ZSM-5 and Zn-Ni/ZSM-5 were calcined at 500 C for 7hr as described in experimental part. Please explain why there was weight loss before 500C in Figure 6.

Reply: The catalyst synthesis process is explained in the methodology section. It describes the impregnation of metals using solutions of metal salts, which is followed by their subsequent calcination to complete the synthesis. On the other hand, the behavior of the catalyst obtained under heating conditions was evaluated using TGA. It shows the amount of volatile material and fixed carbon that the sample can have.

It is important to mention that the scale in Figure 6 is in milligrams (mg), so the mass change is very small.

It has been included: “In this way, both synthesized catalysts exhibit three specific weight loss regions under thermal treatment, as shown in Figure 6. The first weight loss region below 200°C is attributed to physisorbed water molecules, resulting from hydration effects during the catalyst synthesis stage, desorption of organic compounds, and removal of impurities. The second region of weight loss (300-500°C), corresponds to strongly bound water molecules present in the framework structure. The third region at temperatures above 550°C can be attributed to the formation of metal oxides and changes in the crystalline structure of the zeolite [41,42]”.

4) Figure 7a: The data for ZSM-5 is missing. The authors mentioned the reactor was heated up to a temperature of 450C with a heating rate of 15C min-1. However, the results show the reactor reaches 450C in 70min. Please explain the inconsistency.

Reply:  Figure 7a has been modified to include the data of the ZSM-5 catalyst.

On the other hand, the pyrolytic reactor operates with three temperature zones:

  • Jacket temperature, which is used to heat the content within the system.
  • Reactor temperature, to which the recycled plastic waste is exposed and its heating rate (15°C/min) is automatically controlled by the equipment.
  • Exit temperature of the reactor, where the pyrolytic gases leave the reactor and interact with each of the studied catalysts as shown in Figure 1.

Thus, Figure 7a corresponds to the third zone mentioned above, which was monitored by means of the thermocouple coupled to the system.

5) The authors claimed that the catalytic action causes a decrease in the time needed to reach the reaction temperature of 450° C for the reaction without the presence of a catalyst. On the other hand, a slight decrease in time can be observed in the curve corresponding to the bimetallic catalyst. This can be attributed to the slightly higher acidity of the bimetallic catalyst. This is not true in Figure 7a. Also, the temperatures at 0 min was different for two cases (100C and 125C for monometallic and bimetallic one, respectively).

Reply: We appreciate your accurate observation. The modification to figure 7a, as suggested, has been successfully made. However, we have observed a slight difference in the slope of the graph at a temperature of 350°C, indicating a more favorable performance of the bimetallic catalyst.

In addition, it has been included: “On the other hand, a slight decrease in time can be observed in the curve corresponding to the bimetallic catalyst, at temperature about 350°C. This variation can be attributed to the slightly higher acidity of the catalyst, which may have contributed to its enhance performance”.

  • Please improve the quality of the figures (3, 6 and 7) which are blurred 

Reply: It has been modificated.

Round 2

Reviewer 3 Report

In response to the reviewer, the authors write: "Thank you for your appreciation. This was included, "However, in future work, other catalyst characterization techniques such as: i) powder X-ray diffraction (PXRD) for crystal structure determination and identification of phases present in the sample, and ii) inductively coupled plasma (ICP) or X-ray fluorescence (XRF) for more accurate quantitative elemental analysis may be used."

It is worth noting that these methods should be present in the work under review so that future work will not contradict the data presented.

The authors have tried to respond to most of the comments, on the basis of which this paper can be recommended for publication